# SLOWFAST-LLAVA: A STRONG TRAINING-FREE BASELINE FOR VIDEO LARGE LANGUAGE MODELS

## ABSTRACT

We propose SlowFast-LLaVA (or SF-LLaVA for short), a training-free video large language model (LLM) that can jointly capture detailed spatial semantics and long-range temporal context without exceeding the token budget of commonly used LLMs. This is realized by using a two-stream SlowFast design of inputs for Video LLMs to aggregate features from sampled frames in an effective way. Specifically, the Slow pathway extracts features at a low frame rate while keeping as much spatial detail as possible (*e.g.,* with $12 \times 24$ tokens), and the Fast pathway operates on a high frame rate but uses a larger spatial pooling stride (*e.g.,* downsampling $6\times$) to focus on the motion cues. As a result, this design allows us to adequately capture both spatial and temporal features that are beneficial for detailed video understanding. Experimental results show that SF-LLaVA outperforms existing training-free methods on a wide range of video tasks. On some benchmarks, it achieves comparable or even better performance compared to state-of-the-art Video LLMs that are fine-tuned on video datasets.

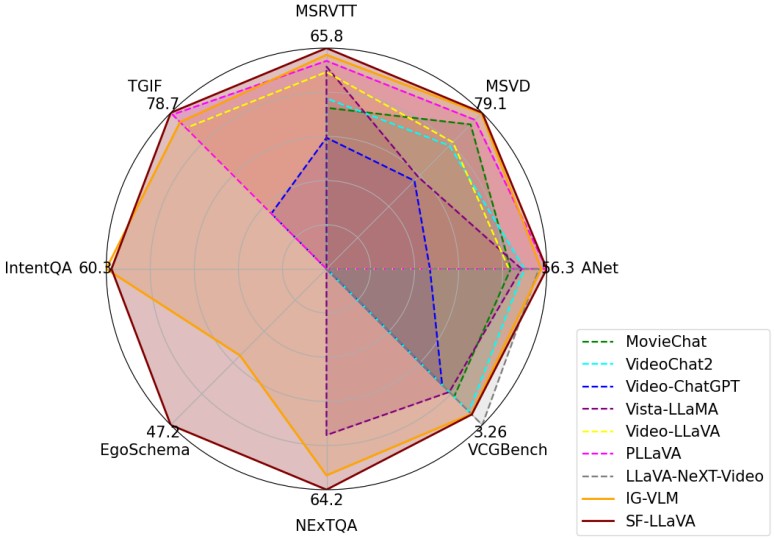

Figure 1: *Comparison with state-of-the-art 7B Video LLMs on 8 video benchmarks.* Training-free and supervised fine-tuned (SFT) Video LLMs are marked using solid (—) and dashed (- - -) lines, respectively. SF-LLaVA outperforms existing training-free methods on all benchmarks, and achieves even better results compared to most SFT methods that are fine-tuned on video datasets.

## 1 INTRODUCTION

Video large language models (LLMs) process video inputs and generate coherent and contextually appropriate responses to user commands by using a pre-trained LLM (Achiam et al., 2023; Chiang et al., 2023; Touvron et al., 2023b; Jiang et al., 2024). Although achieving convincing results, most Video LLMs (Maaz et al., 2024b; Lin et al., 2023; Xu et al., 2024; Zhang et al., 2024b) are fine-tuned on large-scale labeled video datasets, leading to high computational and labeling cost. Recently, training-free methods (Kim et al., 2024; Wu, 2024; Zhang et al., 2024b) have been proposed as a simple and highly cost-efficient solution. They directly use well-trained Image LLMs for video tasks

without additional fine-tuning and demonstrate encouraging performance. However, most existing Video LLMs have two main drawbacks: (1) they work effectively only with a limited number of frames as inputs (*e.g.,* 6 for IG-VLM (Kim et al., 2024) and 16 for PLLaVA (Xu et al., 2024)), making them difficult to capture fine-grained spatial and temporal content throughout the video, and (2) they simply feed the video features into an LLM without a proper temporal modeling design and fully rely on the capability of the LLM to model the motion patterns.

We present **S**low**F**ast-LLaVA (or SF-LLaVA for short), a training-free Video LLM that is built upon LLaVA-NeXT (Liu et al., 2024) without further fine-tuning. Inspired by the successful two-stream networks (Simonyan & Zisserman, 2014; Feichtenhofer et al., 2019) for action recognition, we propose a new SlowFast design of inputs for Video LLMs to capture both detailed spatial semantics and long-range temporal context. Specifically, the Slow pathway extracts features at a low frame rate while keeping spatial information at a higher resolution (*e.g.,* 8 frames each with $24 \times 24$ tokens), and the Fast pathway operates on a high frame rate but uses an aggressive spatial pooling stride (*e.g.,* downsampling each frame to $4 \times 4$ tokens) to focus on motion cues. SF-LLaVA combines the "Slow and Fast features" together as an effective video representation for various tasks. SF-LLaVA has two main advantages over prior work. First, it integrates complementary features from the slowly changing visual semantics and rapidly changing motion dynamics, providing a comprehensive understanding of videos. Second, the dual-pathway design balances the modeling capability and computational efficiency, and enables us to input more video frames to preserve adequate details.

SF-LLaVA takes a video as input by uniformly sampling a large number of frames (denoted as $N$) to maintain as much detail as possible. Frame features $\mathbf{F}_v$ are extracted independently via a visual encoder (*e.g.,* CLIP-L (Radford et al., 2021)) followed by a visual-language adaptor for feature alignment. Then, the features $\mathbf{F}_v$ are fed into the Slow and Fast pathways separately. The Slow pathway uniformly samples $N^{\text{slow}} \ll N$ features from $\mathbf{F}_v$. Prior work (Xu et al., 2024) found that properly pooling frame features can improve both efficiency and robustness. We follow them to aggregate features in the Slow pathway by using a pooling with a small stride (*e.g.,* $1 \times 2$) over the spatial dimensions. The Fast pathway takes all $N^{\text{fast}} = N$ features and performs a more aggressive spatial pooling of each frame to focus on a finer temporal resolution. Finally, visual tokens from both pathways are concatenated and fed into the LLM to generate the answer.

We extensively evaluate SF-LLaVA on 3 video tasks (*i.e.,* Open-Ended VideoQA, Multiple Choice VideoQA, and Text Generation) with 8 benchmark, including videos from various types (*e.g.,* first- and third-person views) and lengths (*e.g.,* short and long videos). Experimental results (as shown in Fig. 1) show that SF-LLaVA outperforms existing training-free methods by a clear margin on all benchmarks, and achieves on-par or even better performance compared to SFT models that have been carefully fine-tuned on video datasets. We also conduct comprehensive ablation studies on our SlowFast design recipe, which hopefully provide some valuable insights for future work.

## 2 RELATED WORK

**Image Large Language Models.** Significant advances have been observed in the development of multimodal large language models (LLMs) (Achiam et al., 2023; Team et al., 2023; McKinzie et al., 2024; Abdin et al., 2024; Liu et al., 2024). As a pioneer work, Flamingo (Alayrac et al., 2022) accepts arbitrarily interleaved visual and text data as inputs and generates text in an open-ended manner. BLIP-2 (Li et al., 2023b) uses pre-trained visual and text models, and bridges the domain gap with the proposed Q-Former. LLaVA(-v1.5/NeXT) (Liu et al., 2023b;a; 2024) achieves remarkable performance by leveraging a simple linear connector or an MLP between visual and text models and designing an efficient instruction following data pipeline assisted with GPT. More recently, MM1 (McKinzie et al., 2024) conducts comprehensive ablation studies on model components and data choices, and offers valuable insights for understanding Image LLMs. There are also efforts to ingest other modalities. Ferret (You et al., 2023; Zhang et al., 2024a) focuses on the box/shape modality and enhances a model's language grounding capability at any granularity. 3D-LLM (Hong et al., 2023) enables open-ended question answering in 3D by injecting 3D representations into an LLM. 4M (Mizrahi et al., 2023; Bachmann et al., 2024) presents a general any (modality) to any (modality) framework with strong out-of-box perceptual and generative capabilities.

**Video Large Language Models.** With the rapid development of LLMs (Achiam et al., 2023; Team et al., 2023; Chiang et al., 2023; Touvron et al., 2023a;b), there is increasing interest in generalist video models that can perform a wide range of video tasks. Video-ChatGPT (Maaz et al., 2024b)

extracts per-frame features then aggregates them by using two spatial and temporal pooling operations before inputting them to an LLM. VideoChat (Li et al., 2023c) encodes a video as both video text descriptions and video appearance embeddings. Video-LLaVA (Lin et al., 2023) pre-aligns the image and video encoders, and learns a shared projector to project them to the language space. PLLaVA (Xu et al., 2024) achieves convincing performance by fine-tuning a pre-trained Image LLM on video understanding data. LLaVA-NeXT-Video (Zhang et al., 2024b) improves LLaVA-NeXT (Liu et al., 2024) by fine-tuning it on video data, and its DPO version (Zhang et al., 2024b) further aligns the model responses with AI feedbacks.

**Training-Free Video LLMs** are built upon Image LLMs and require no additional fine-tuning to work for video scenarios. FreeVA (Wu, 2024) explores different temporal aggregation methods and effectively pools video features before sending them to an LLM. IG-VLM (Kim et al., 2024) assembles multiple video frames into an image grid and uses the Image LLM as it is on the image grid for video tasks. These training-free models show encouraging results on various benchmarks, but they have two main drawbacks. First, they can only successfully process a few frames from a video (*e.g.,* 4 frames in FreeVA and 6 frames in IG-VLM), which limits them to work only for short and simple videos. Second, they simply ingest the video features and fully rely on the capability of the LLMs to capture the temporal dependency along the video. In this paper, we propose a new SlowFast design to capture both detailed spatial and temporal cues for video understanding by effectively and efficiently taking more frames (*e.g.,* 50) as inputs.

## 3 SLOWFAST-LLAVA

We introduce a training-free Video LLM, named **S**low**F**ast-LLaVA (or SF-LLaVA for short), based on the LLaVA-NeXT (Liu et al., 2024), as shown in Fig. 2. Inspired by (Simonyan & Zisserman, 2014; Feichtenhofer et al., 2019) for action recognition, we propose a SlowFast design that uses two-stream inputs for Video LLMs to jointly capture detailed spatial semantic and long-range temporal context without exceeding the token budget of commonly used LLMs. (*e.g.,* 4096 in Vicuna-v1.5). Specifically, the Slow pathway includes "high-resolution"[1] but low-frame-rate frame features (*e.g.,* 8 frames each with $12 \times 24$ tokens) to capture spatial detail as much as possible, and the Fast pathway includes "low-resolution" but high-frame-rate frame features (*e.g.,* 64 frames each with $4 \times 4$ tokens) to model greater temporal context. This design allows us to adequately preserve both spatial and temporal information, and aggregate them together as a powerful video representation.

### 3.1 PRELIMINARIES: TRAINING-FREE VIDEO LLMS

A training-free Video LLM is built upon a pre-trained Image LLM *without further fine-tuning on any data*. It saves significant computation resources and model training time, and offers a greater flexibility that can be quickly adapted into different application scenarios. The main effort of this research direction is to improve the visual representation (*e.g.,* organizing sampled frames (Kim et al., 2024) or incorporating textual descriptions (Zhang et al., 2023a)) and effectively leveraging the knowledge of a pre-trained LLM to better fit into the video tasks.

Given a video $\mathbf{V}$, a frame sampler first selects $N$ key frames (denoted as $\mathbf{I}$).[2] The sampled frames are either arranged to a combined image grid (Kim et al., 2024) or treated independently (Wu, 2024; Zhang et al., 2024b) as the inputs to the model. Video features are extracted as $\mathbf{F}_v = \texttt{Visual}_{\texttt{enc}}(\mathbf{I})$, where $\texttt{Visual}_{\texttt{enc}}$ is an image-based visual encoder, such as CLIP-L (Radford et al., 2021).[3] Note that IG-VLM (Kim et al., 2024) uses the AnyRes (Liu et al., 2024) technique to extract features from a combined image grid, and most other methods, such as FreeVA (Wu, 2024), extracts features from each frame independently. Before inputting the video features $\mathbf{F}_v$ into the LLM, a feature aggregator, $\mathbf{F}_v^{\text{aggr}} = \texttt{Aggregator}(\mathbf{F}_v)$, is usually used to aggregate visual features using pre-defined pooling operations. This stage aims to (1) leverage the temporal prior knowledge for better video representation and (2) reduce the number of video tokens to avoid exceeding the LLM's token limit. Finally, the aggregated video features $\mathbf{F}_v^{\text{aggr}}$ and the question $\mathbf{Q}$ are fed into the LLM to get a

---

[1]We mean "high- or low-resolution" frames by their number of tokens after the visual encoder and pooling, such as $24 \times 24$ or $4 \times 4$, not the raw image size. We extract features for all frames in size of $336 \times 336$.

[2]Most existing methods uniformly sample frames from a video for both effectiveness and simplicity.

[3]An Image LLM usually has a projector, such as MLPs, between its visual encoder and the LLM to align the visual and text modalities. Unless noted otherwise, we extract the features after the projector.

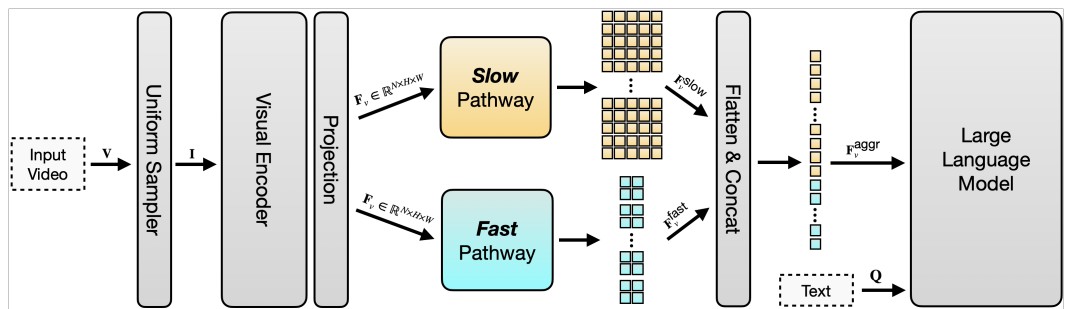

Figure 2: *Visualization of SlowFast-LLaVA*, which is a training-free model built upon LLaVA-NeXT without further fine-tuning. The Slow pathway (in color yellow) extracts features at a low frame rate while keeping as much spatial detail as possible with more tokens, and the Fast pathway (in color blue) operates on a high frame rate but applies a larger spatial pooling stride to focus on the motion cue. This design allows us to adequately preserve adequate spatial and temporal information, and aggregate them together as an effective representation for detailed video understanding.

corresponding answer, as shown in Eq. 1.

$$\mathbf{A} = \texttt{LLM}(\mathbf{Prompt}, \texttt{Aggregator}(\texttt{Visual}_{\text{enc}}(\mathbf{I})), \mathbf{Q}), \tag{1}$$

where $\mathbf{Prompt}$ denotes the system prompt or the instruction that is used to properly guide an LLM for obtaining desirable answers. Since training-free Video LLMs directly use an image-based vision-language model (VLM) for video understanding, it is essential to modify the original prompt to accommodate the change from image to video scenarios. We will experiment for different prompts and show the importance of using a proper instruction design for Video LLMs in Sec. 4.5.

## 3.2 SLOWFAST ARCHITECTURE

As shown in Fig. 2, our SF-LLaVA follows the standard training-free Video LLM pipeline. It takes a video $\mathbf{V}$ and a question $\mathbf{Q}$ as inputs, and outputs an answer $\mathbf{A}$, in response to $\mathbf{Q}$. For the input, we uniformly sample $N$ frames, $\mathbf{I} = \{I_1, I_2, ..., I_N\}$, from each video in an arbitrary size and length, without special frame assembling. The video features are extracted frame by frame independently as $\mathbf{F}_v \in \mathbb{R}^{N \times H \times W}$, where $H$ and $W$ are the height and width of the frame feature. Then, we further process $\mathbf{F}_v$ in two streams (*i.e.,* the Slow and Fast pathways as follows), and combine them together as an effective video representation.

**The Slow pathway** uniformly samples $N^{\text{slow}}$ frame features from $\mathbf{F}_v$, where $N^{\text{slow}} \ll N$ since it operates on a low frame rate. Since prior work (Xu et al., 2024) found that pooling "properly" (*e.g.,* stride $2 \times 2$) along the spatial dimension improves both the efficiency and robustness, we reserve the opportunity to apply the pooling over $\mathbf{F}_v$ with stride $\sigma_h \times \sigma_w$ and gets the final feature $\mathbf{F}_v^{\text{slow}} \in \mathbb{R}^{N^{\text{slow}} \times H^{\text{slow}} \times W^{\text{slow}}}$, where $H^{\text{slow}} = H/\sigma_h$ and $W^{\text{slow}} = W/\sigma_w$. The whole process of the Slow pathway can be summarized in Eq. 2.

$$\mathbf{F}_v \in \mathbb{R}^{N \times H \times W} \xrightarrow[\text{temporal downsample}]{\text{spatial pool}} \mathbf{F}_v^{\text{slow}} \in \mathbb{R}^{N^{\text{slow}} \times H^{\text{slow}} \times W^{\text{slow}}} \tag{2}$$

**The Fast pathway** keeps all frame features from $\mathbf{F}_v$ to capture temporal context as much as possible along the video. Specifically, we aggressively downsample $\mathbf{F}_v$ with a large spatial pooling stride $\gamma_h \times \gamma_w$ and gets the final feature $\mathbb{R}^{N^{\text{fast}} \times H^{\text{fast}} \times W^{\text{fast}}}$, where $N^{\text{fast}} = N$, $H^{\text{fast}} = H/\gamma_h$, and $W^{\text{fast}} = W/\gamma_w$. We set $H^{\text{fast}} \ll H$ and $W^{\text{fast}} \ll W$ to make the Fast pathway to focus on modeling the temporal context and motion cues. Formally, the whole process of the Fast pathway is as in Eq. 3.

$$\mathbf{F}_v \in \mathbb{R}^{N \times H \times W} \xrightarrow{\text{spatial pool}} \mathbf{F}_v^{\text{fast}} \in \mathbb{R}^{N^{\text{fast}} \times H^{\text{fast}} \times W^{\text{fast}}}, \text{ where } N^{\text{fast}} = N \tag{3}$$

Finally, the aggregated video feature is obtained by $\mathbf{F}_v^{\text{aggr}} = [\texttt{flat}(\mathbf{F}_v^{\text{slow}}), \texttt{flat}(\mathbf{F}_v^{\text{fast}})]$, where $\texttt{flat}$ and $[,]$ indicate the flatten and concatenation operations, respectively. As the equation implies, we do not use any special tokens in $\mathbf{F}_v^{\text{aggr}}$ to separate the Slow and Fast pathways. Thus, SF-LLaVA uses $N^{\text{slow}} \times H^{\text{slow}} \times W^{\text{slow}} + N^{\text{fast}} \times H^{\text{fast}} \times W^{\text{fast}}$ video tokens in total.

The visual features $\mathbf{F}_v^{\text{aggr}}$ will be concatenated with the text tokens (including both prompt and question) as the inputs to the LLM as in Eq. 1. An overview of our SlowFast pipeline is summarized as in Eq. 4, where `Slow` and `Fast` indicate our Slow and Fast aggregation pipelines as above.

$$\mathbf{A} = \text{LLM}(\mathbf{Prompt}, [\text{Slow}(\mathbf{F}_v), \text{Fast}(\mathbf{F}_v)], \mathbf{Q}), \text{ where } \mathbf{F}_v = \text{Visual}_{\text{enc}}(\mathbf{I}) \qquad (4)$$

## 4 EXPERIMENTS

### 4.1 BENCHMARKS AND METRICS

**Open-Ended VideoQA** expects the model to generate answers in freestyle in response to a question for a video. We include MSVD-QA (Chen & Dolan, 2011), MSRVTT-QA (Xu et al., 2016), TGIF-QA (Li et al., 2016) and ActivityNet-QA (or ANet-QA in tables) (Yu et al., 2019) as the benchmarks for this task. Except for ActivityNet-QA, we follow prior work (Maaz et al., 2024b) and report the performance on the validation set. We use the GPT-assisted evaluation to assess the accuracy (accuracy with the answer being true or false) and the quality (score ranging from 0 to 5) of the models. As pointed out by FreeVA (Wu, 2024) different GPT versions can significantly impact the results, we report to use `GPT-3.5-Turbo-0125` to perform a fair comparison.

**Multiple Choice VideoQA** presents a set of multiple choice options to Video LLMs and evaluates their capability of picking the correct choice. Specifically, we evaluate our model on NExTQA (Xiao et al., 2021), EgoSchema (Mangalam et al., 2024) and IntentQA (Li et al., 2023a). The accuracy of selecting the correct answer from the options is used as the evaluation metric.

**Text Generation** is used to evaluate the text generation performance of a Video LLM, and especially focuses on the following aspects: Correctness of Information (CI), Detail Orientation (DO), Contextual Understanding (CU), Temporal Understanding (TU), and Consistency (CO). We use the VCGBench (Maaz et al., 2024b) to evaluate these tasks and follow its official pipeline to evaluate this capability. Specifically, we use `GPT-3.5-Turbo-0125` for evaluation.

### 4.2 IMPLEMENTATION DETAILS

**Experimental Settings.** We perform all experiments on a system with 8 Nvidia A100 80G graphics cards. SF-LLaVA is built upon LLaVA-NeXT (Liu et al., 2024) 7B and 34B models. We use their pre-trained weights available on HuggingFace[4]. To deal with long sequences, we follow LLaVA-NeXT-Video (Zhang et al., 2024b) to apply the rotary position embedding (RoPE) (Su et al., 2024), and use the scaling factor of 2, which doubles the context length to 8192 tokens.

**Input and Model Settings.** SF-LLaVA takes as inputs a video with arbitrary size and length, and uniformly samples $N = 50$ frames as key frames. The key frames are resized to $336 \times 336$, and the visual encoder (*i.e.,* OpenAI's CLIP-L-14) will output $24 \times 24$ tokens for each of them. For the Slow pathway, we uniformly select $N^{\text{slow}} = 10$ frame features from $\mathbf{F}_v$ and pool their extracted features to $10 \times 12 \times 24$; for the Fast pathway, we use features of all frames (*i.e.,* $N^{\text{fast}} = N = 50$) and pool their extracted features to $50 \times 4 \times 4$. Thus, SF-LLaVA uses $10 \times 12 \times 24 + 50 \times 4 \times 4 = 3680$ visual tokens in total, and we choose this as the maximum number since the inference on the SF-LLaVA-34B model already reaches 80G GPU memory. The SlowFast video tokens are then concatenated with the text tokens as inputs to the LLM.

### 4.3 MAIN RESULTS

**Open-Ended VideoQA** results are shown in Table 1. SF-LLaVA obtains better performance than existing training-free methods on all benchmarks. Specifically, SF-LLaVA outperforms IG-VLM (Kim et al., 2024) by 2.1% and 5.0% on MSRVTT-QA, 5.7% and 1.5% on TGIF-QA, 1.2% and 0.8% on ActivityNet-QA, using 7B and 34B LLMs, respectively. When even compared to state-of-the-art SFT methods, SF-LLaVA achieves on-par results on most benchmarks (*i.e.,* MSVD-QA, MSRVTT-QA, and TGIF-QA), and only the results of PLLaVA (Xu et al., 2024) and LLaVA-NeXT-Video-DPO (Zhang et al., 2024b) are better than ours on ActivityNet-QA.

**Multiple Choice VideoQA** results are shown in Table 2. SF-LLaVA outperforms other training-free methods that use comparable LLMs and visual encoders, such as IG-VLM (Kim et al., 2024)

---

[4] https://huggingface.co/collections/liuhaotian/llava-16-65b9e40155f60fd046a5ccf2

| Method | LLM Size | Vision Encoder | Open-Ended VideoQA (Accuracy/Score) | | | |
|---|---|---|---|---|---|---|
| | | | MSVD-QA | MSRVTT-QA | TGIF-QA | ANet-QA |
| Video-LLaMA (Zhang et al., 2023b) | 7B | CLIP-G | 51.6/2.5 | 29.6/1.8 | - | 12.4/1.1 |
| Video-LLaMA2 (Cheng et al., 2024) | 7B | CLIP-L | 70.9/3.8 | - | - | 50.2/3.3 |
| Video-ChatGPT (Maaz et al., 2024b) | 7B | CLIP-L | 64.9/3.3 | 49.3/2.8 | 51.4/3.0 | 35.2/2.7 |
| VideoGPT+ (Maaz et al., 2024a) | 3.8B | CLIP-L | 72.4/3.9 | 60.6/3.6 | 74.6/4.1 | 50.6/3.6 |
| Video-LLaVA (Lin et al., 2023) | 7B | ViT-L | 70.7/3.9 | 59.2/3.5 | 70.0/4.0 | 45.3/3.3 |
| MovieChat (Song et al., 2023) | 7B | CLIP-G | 75.2/3.8 | 52.7/2.6 | - | 45.7/3.4 |
| MovieChat+ (Song et al., 2024) | 7B | CLIP-G | 76.5/3.9 | 53.9/2.7 | - | 48.1/3.4 |
| VideoChat (Li et al., 2023c) | 7B | CLIP-G | 56.3/2.8 | 45.0/2.5 | 34.4/2.3 | 26.5/2.2 |
| VideoChat2 (Li et al., 2023d) | 7B | UMT-L | 70.0/3.9 | 54.1/3.3 | - | 49.1/3.3 |
| Vista-LLaMA (Ma et al., 2023) | 7B | CLIP-G | 65.3/3.6 | 60.5/3.3 | - | 48.3/3.3 |
| LLaMA-VID (Li et al., 2023e) | 13B | CLIP-G | 69.7/3.7 | 57.7/3.2 | - | 47.4/3.3 |
| PLLaVA (Xu et al., 2024) | 7B | CLIP-L | 76.6/4.1 | 62.0/3.5 | 77.5/4.1 | 56.3/3.5 |
| LLaVA-NeXT-Video (Zhang et al., 2024b) | 7B | CLIP-L | - | - | - | 53.5/3.2 |
| LLaVA-NeXT-Video-DPO (Zhang et al., 2024b) | 7B | CLIP-L | - | - | - | 60.2/3.5 |
| FreeVA (Wu, 2024) | 7B | CLIP-L | 73.8/4.1 | 60.0/3.5 | - | 51.2/3.5 |
| DeepStack-L (Meng et al., 2024) | 7B | CLIP-L | 76.0/4.0 | - | - | 49.3/3.1 |
| LLaVA-NeXT-Image (Zhang et al., 2024b) | 7B | CLIP-L | - | - | - | 53.8/3.2 |
| IG-VLM (LLaVA-v1.6) (Kim et al., 2024) | 7B | CLIP-L | 78.8/4.1 | 63.7/3.5 | 73.0/4.0 | 54.3/3.4 |
| **SF-LLaVA-7B** | 7B | CLIP-L | **79.1/4.1** | **65.8/3.6** | **78.7/4.2** | **55.5**/3.4 |

(a) *All models use 7B or comparable LLMs*. SF-LLaVA outperforms state-of-the-art training-free methods by 0.3% on MSVD-QA, 2.1% on MSRVTT-QA, 5.7% on TGIF-QA, and 2.0% on ANet-QA. SF-LLaVA also achieves better performance than most SFT methods on these benchmarks.

| Method | LLM Size | Vision Encoder | Open-Ended VideoQA (Accuracy/Score) | | | |
|---|---|---|---|---|---|---|
| | | | MSVD-QA | MSRVTT-QA | TGIF-QA | ANet-QA |
| Video-LLaMA2 (Cheng et al., 2024) | 46.7B | CLIP-L | 70.5/3.8 | | | 50.3/3.4 |
| PLLaVA (Xu et al., 2024) | 34B | CLIP-L | 79.9/4.2 | 68.7/3.8 | 80.6/4.3 | 60.9/3.7 |
| LLaVA-NeXT-Video (Zhang et al., 2024b) | 34B | CLIP-L | - | - | - | 58.8/3.4 |
| LLaVA-NeXT-Video-DPO (Zhang et al., 2024b) | 34B | CLIP-L | - | - | - | 64.4/3.6 |
| LLaVA-NeXT-Image (Zhang et al., 2024b) | 34B | CLIP-L | - | - | - | 55.6/3.3 |
| IG-VLM (LLaVA-v1.6) (Kim et al., 2024) | 34B | CLIP-L | 79.6/4.1 | 62.4/3.5 | 79.1/4.2 | 58.4/3.5 |
| **SF-LLaVA-34B** | 34B | CLIP-L | **79.9/4.1** | **67.4/3.7** | **80.6/4.3** | **59.2**/3.5 |

(b) *All models use 34B or stronger LLMs*. SF-LLaVA outperforms state-of-the-art training-free methods by 0.3% on MSVD-QA, 5.0% on MSRVTT-QA, 1.5% on TGIF-QA, and 0.8% on ANet-QA.

Table 1: *Open-Ended VideoQA results.* **Bold numbers** are the best among training-free methods and underlined numbers are the best among all Video LLMs. Methods below the dashed line (- - -) are the training-free baselines, and others are models fine-tuned on additional video data.

on all benchmarks. Specifically, on the challenging EgoSchema dataset, which involves complex long-form temporal reasoning (Mangalam et al., 2024), SF-LLaVA outperforms IG-VLM by 11.4% and 2.2% when using 7B and 34B LLMs, respectively. This highlights the ability of SF-LLaVA on long-form video understanding. Note that VideoTree (Wang et al., 2024b) is leading the benchmark because it is built upon a proprietary LLM (*i.e.,* GPT-4 (Achiam et al., 2023)) whose performance is much better than the open-sourced LLMs. When compared to SFT methods (Cheng et al., 2024), SF-LLaVA 34B model also achieves better results (+2.5%) on EgoSchema, which confirms the capability of our SlowFast design on long videos.

**Text Generation** benchmarks are shown in Table 3, where SF-LLaVA-34B outperforms all training-free baselines on average. First, we observe that SF-LLaVA consistently performs worse than LLaVA-NeXT-Image (Zhang et al., 2024b) on Detail Orientation (DO). This is because LLaVA-NeXT-Image takes more "high-resolution" input frames than ours (*i.e.,* 32 frames with $12 \times 12$ tokens *v.s.* 10 frames with $12 \times 24$ tokens), thus is able to capture more spatial information. Second, SF-LLaVA takes advantage of the SlowFast design to cover a longer temporal context by using even fewer visual tokens (*i.e.,* 4608 tokens *v.s.* 3680 tokens), thus excels in all other tasks, especially in Temporal Understanding (TU). Third, we observe that SF-LLaVA-34B is superior to most SFT methods (*e.g,* outperforming Video-LLaMA2 (Cheng et al., 2024) +0.1 score on TU and +0.31 score on CO), but only needs to catch up with LLaVA-NeXT-Video-DPO (Zhang et al., 2024b).

## 4.4 Design Choices of SlowFast

We first validate if both the Slow and Fast pathways are essential, and continue to experiment for their design choices respectively. These ablation studies are conducted on ActivityNet-QA (an Open-Ended VideoQA dataset that contains videos of human activities) and EgoSchema (a Multiple Choice VideoQA dataset requiring long-form understanding of egocentric videos).

| Method | LLM Size | Vision Encoder | Multiple Choice VideoQA (Accuracy) | | |
|---|---|---|---|---|---|
| | | | NExTQA | EgoSchema | IntentQA |
| Video-LLaMA2 (Cheng et al., 2024) | 7B | CLIP-L | - | 51.7 | - |
| MovieChat+ (Song et al., 2024) | 7B | CLIP-G | 54.8 | - | - |
| Vista-LLaMA (Ma et al., 2023) | 7B | CLIP-G | 60.7 | - | - |
| DeepStack-L (Meng et al., 2024) | 7B | CLIP-L | 61.0 | 38.4 | - |
| IG-VLM (LLaVA-v1.6) (Kim et al., 2024) | 7B | CLIP-L | 63.1 | 35.8 | **60.3** |
| **SF-LLaVA-7B** | 7B | CLIP-L | **64.2** | **47.2** | 60.1 |

(a) *All models use 7B or comparable LLMs.* SF-LLaVA outperforms state-of-the-art training-free methods by 1.1% on NExTQA and 11.4% on EgoSchema.

| Method | LLM Size | Vision Encoder | Multiple Choice VideoQA (Accuracy) | | |
|---|---|---|---|---|---|
| | | | NExTQA | EgoSchema | IntentQA |
| Video-LLaMA2 (Cheng et al., 2024) | 46.7B | CLIP-L | - | 53.3 | - |
| LLoVi (Zhang et al., 2023a) | GPT-3.5 | Unknown | 67.7 | 50.3 | 64.0 |
| VideoAgent (Wang et al., 2024a) | GPT-4 | Unknown | 71.3 | 60.2 | - |
| VideoTree (Wang et al., 2024b) | GPT-4 | Unknown | 73.5 | 66.2 | 66.9 |
| IG-VLM (LLaVA-v1.6) (Kim et al., 2024) | 34B | CLIP-L | 70.9 | 53.6 | 65.3 |
| **SF-LLaVA-34B** | 34B | CLIP-L | 72.0 | 55.8 | 66.5 |

(b) *All models use 34B or stronger LLMs.* VideoAgent and VideoTree use proprietary stronger LLMs, thus win the leaderboard. On the other hand, SF-LLaVA outperforms baselines using comparable LLMs.

Table 2: *Multiple Choice VideoQA results.* **Bold numbers** are the best among training-free methods and underlined numbers are the best among all Video LLMs. Methods below the dashed line (- - -) are the training-free baselines, and others are SFT methods fine-tuned by massive video data.

| Method | LLM Size | Vision Encoder | Text Generation (Score) | | | | | |
|---|---|---|---|---|---|---|---|---|
| | | | CI | DO | CU | TU | CO | Average |
| Video-LLaMA (Zhang et al., 2023b) | 7B | CLIP-G | 1.96 | 2.18 | 2.16 | 1.82 | 1.79 | 1.98 |
| Video-LLaMA2 (Cheng et al., 2024) | 7B | CLIP-L | 3.16 | 3.08 | 3.69 | 2.56 | 3.14 | 3.13 |
| Video-ChatGPT (Maaz et al., 2024b) | 7B | CLIP-L | 2.50 | 2.57 | 2.69 | 2.16 | 2.20 | 2.42 |
| VideoGPT+ (Maaz et al., 2024a) | 3.8B | CLIP-L | 3.27 | 3.18 | 3.74 | 2.83 | 3.39 | 3.28 |
| MovieChat (Song et al., 2023) | 7B | CLIP-G | 2.76 | 2.93 | 3.01 | 2.24 | 2.42 | 2.67 |
| VideoChat (Li et al., 2023c) | 7B | CLIP-G | 2.23 | 2.50 | 2.53 | 1.94 | 2.24 | 2.29 |
| VideoChat2 (Li et al., 2023d) | 7B | UMT-L | 3.02 | 2.88 | 3.51 | 2.66 | 2.81 | 2.98 |
| Vista-LLaMA (Ma et al., 2023) | 7B | CLIP-G | 2.44 | 2.64 | 3.18 | 2.26 | 2.31 | 2.57 |
| LLaMA-VID (Li et al., 2023e) | 13B | CLIP-G | 2.96 | 3.00 | 3.53 | 2.46 | 2.51 | 2.89 |
| LLaVA-NeXT-Video (Zhang et al., 2024b) | 7B | CLIP-L | 3.39 | 3.29 | 3.92 | 2.60 | 3.12 | 3.26 |
| LLaVA-NeXT-Video-DPO (Zhang et al., 2024b) | 7B | CLIP-L | 3.64 | 3.45 | 4.17 | 2.95 | 4.08 | 3.66 |
| LLaVA-NeXT-Image (Zhang et al., 2024b) | 7B | CLIP-L | 3.05 | **3.12** | **3.68** | 2.37 | 3.16 | **3.07** |
| IG-VLM (LLaVA-v1.6) (Kim et al., 2024) | 7B | CLIP-L | **3.11** | 2.78 | 3.51 | 2.44 | 3.29 | 3.03 |
| **SF-LLaVA-7B** | 7B | CLIP-L | 3.09 | 2.70 | 3.57 | **2.52** | **3.35** | 3.04 |

(a) *All models use 7B or comparable LLMs.* SF-LLaVA is leading the Temporal Understanding (TU) benchmark, which confirms the capability of our SlowFast design on modeling temporal context.

| Method | LLM Size | Vision Encoder | Text Generation (Score) | | | | | |
|---|---|---|---|---|---|---|---|---|
| | | | CI | DO | CU | TU | CO | Average |
| Video-LLaMA2 (Cheng et al., 2024) | 46.7B | CLIP-L | 3.08 | 3.11 | 3.64 | 2.67 | 3.26 | 3.15 |
| LLaVA-NeXT-Video (Zhang et al., 2024b) | 34B | CLIP-L | 3.48 | 3.37 | 3.95 | 2.64 | 3.28 | 3.34 |
| LLaVA-NeXT-Video-DPO (Zhang et al., 2024b) | 34B | CLIP-L | 3.81 | 3.55 | 4.24 | 3.14 | 4.12 | 3.77 |
| LLaVA-NeXT-Image (Zhang et al., 2024b) | 34B | CLIP-L | 3.29 | **3.23** | 3.83 | 2.51 | 3.47 | 3.27 |
| IG-VLM (LLaVA-v1.6) (Kim et al., 2024) | 34B | CLIP-L | 3.11 | 2.78 | 3.51 | 2.44 | 3.29 | 3.03 |
| **SF-LLaVA-34B** | 34B | CLIP-L | **3.48** | 2.96 | **3.84** | **2.77** | **3.57** | **3.32** |

(b) *All models use 34B or stronger LLMs.* SF-LLaVA outperforms the state-of-the-art training-free method (LLaVA-NeXT-Image) by +0.05 score on average and gets +0.19 score on CI and +0.26 score on TU.

Table 3: *Text Generation results.* **Bold numbers** are the best among training-free methods and underlined numbers are the best among all Video LLMs. Methods below the dashed line (- - -) are the training-free baselines, and others are SFT methods fine-tuned by massive video data.

**Can we remove the Slow pathway?** First, we simply remove the Slow pathway, while keeping the Fast pathway as 50 frames, each with $4 \times 4$ tokens. Fig. 3 shows that, on all benchmarks, removing Slow pathway ($N^{slow}$ equals to 0) will introduce much lower performance. Second, we validate if the performance gain is caused by the necessity of the Slow pathway or the increased visual tokens brought by using more frames. We test this by gradually increasing $N^{fast}$ from 50 to 225 to compensate for the loss of visual tokens. Results in Table 4 show that using larger $N^{fast}$ generally obtains better results, but the results quickly saturate when $N^{fast}$ is larger than 150. We also compare the baselines in Table 4, which use $N^{fast} = 200$, and SF-LLaVA models in Fig. 3 with $N^{slow} = 8$ and $N^{fast} = 50$, since these models use comparable number of tokens (3200 *v.s.* 3104) in total. Results show that SF-LLaVA outperforms this new baseline under all settings (*e.g.,* 54.6%

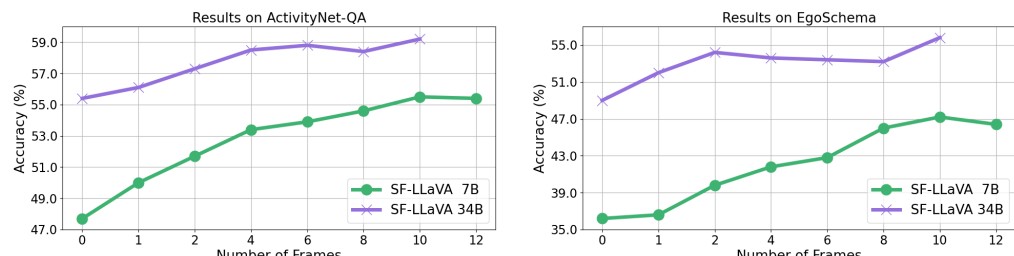

Figure 3: *Effect of using different numbers of frames in the Slow pathway.*

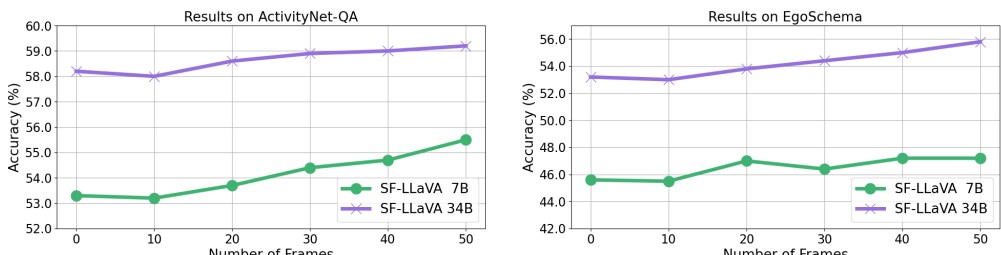

Figure 4: *Effect of using different numbers of frames in the Fast pathway.*

| | | Number of Frames in the Fast Pathway | | | | | |
| | | 100 | 125 | 150 | 175 | 200 | 225 |
| **SF-LLaVA-7B** | ANet-QA | 48.8/3.2 | 49.1/3.2 | 49.6/3.2 | 50.0/3.2 | 49.7/3.2 | 50.0/3.2 |
| | EgoSchema | 36.2 | 37.0 | 36.6 | 36.8 | 37.0 | 38.2 |
| **SF-LLaVA-34B** | ANet-QA | 55.6/3.4 | 55.8/3.4 | 55.5/3.4 | 55.1/3.4 | 55.4/3.4 | – |
| | EgoSchema | 49.8 | 51.4 | 50.6 | 52.2 | 52.0 | – |

Table 4: *Effect of increasing $N^{\text{fast}}$ while keeping $N^{\text{slow}} = 0$. Each frame in the Fast pathway outputs $4 \times 4$ tokens. The symbol "–" means the setting gets out-of-memory on 80GB GPUs.*

*v.s.* 49.7% and 46.0% *v.s.* 37.0% using 7B LLMs on ActivityNet-QA and EgoSchema, respectively). All of the above results demonstrate that it is essential to use the Slow pathway in SF-LLaVA.

**Can we remove the Fast pathway?** We validate this by removing the Fast pathway while retaining the Slow pathway (having 10 frames, each with $12 \times 24$ tokens). Fig. 4 shows that SF-LLaVA with $N^{\text{fast}} = 50$ consistently outperforms this baseline. Similar to the experiments for the Slow pathway, we increase $N^{\text{slow}}$ to ensure SF-LLaVA and this new baseline have a comparable number of input video tokens. Specifically, we increase $N^{\text{slow}}$ to 12 frames, which is the maximum number of frames that the 34B model can afford under 80GB GPU memory. SF-LLaVA still outperforms this baseline on both ActivityNet-QA (55.5% *v.s.* 54.1% on 7B model and 59.2% *v.s.* 58.8% on 34B model) and EgoSchema (47.2% *v.s.* 46.6% on 7B model and 55.8% *v.s.* 54.6% on 34B model). We observe that the performance gap is more significant on EgoSchema, since it mostly contains long-form videos and answering the questions requires capturing longer context using the Fast pathway.

**Pooling impact on Slow pathway.** We analyze the effect of using different pooling strategies over $\mathbf{F}_v^{\text{slow}}$. The Fast pathway is kept as in Sec. 4.2. First, Table 5 (row 1 *v.s.* others) shows that keeping visual tokens as many as possible is a viable way to obtain better results on average, however, to cover longer context, we can easily reach the limits of an LLM's context window and the GPU memory (*e.g.,* the 34B model). Second, pooling properly over either the spatial or temporal dimension (*e.g.,* $2\times$ in row 2 and 4) can also improve the performance (*e.g.,* $\sim$1% on ActivityNet-QA) but using an aggressive pooling can decrease the performance a lot (row 1 *v.s.* row 6). This also matches the observations in prior work (Xu et al., 2024; Wu, 2024), Third, when preserving the same number of tokens (*e.g.,* row 2 and row 4), spatial pooling is better than temporal pooling, especially on the benchmarks (*e.g.,* EgoSchema) that require strong temporal modeling capabilities.

**Number of frames in Slow pathway.** We evaluate the effect of using different numbers of frames $N_s$ in the Slow pathway. In particular, we test $N_s \in \{1, 2, 4, 6, 8, 10\}$ as shown in Fig. 3, while keeping $\mathbf{F}_v^{\text{fast}}$ in size of $50 \times 4 \times 4$. Note that we choose the max length to make sure the GPU memory usage of SF-LLaVA-34B inference is under 80GB. The results show that increasing the number of frames in the Slow pathway can improve the performance on both ActivityNet-QA and EgoSchema. Thus we set $N^{\text{slow}}$ to 10 to achieve the best possible performance of SF-LLaVA.

| Model Size | Output #tokens | ANet-QA | EgoSchema |
|---|---|---|---|
|  | $10 \times 24 \times 24$ | 54.0/3.3 | 53.2 |
|  | $10 \times 12 \times 24$ | 55.5/3.3 | 47.2 |
| **7B** | $10 \times 12 \times 12$ | 54.7/3.3 | 39.0 |
|  | $5 \times 24 \times 24$ | 54.5/3.3 | 44.4 |
|  | $5 \times 12 \times 24$ | 53.9/3.3 | 39.4 |
|  | $5 \times 12 \times 12$ | 51.4/3.2 | 36.4 |

| Model size | Output #tokens | ANet-QA | EgoSchema |
|---|---|---|---|
|  | $10 \times 24 \times 24$ | - | - |
|  | $10 \times 12 \times 24$ | 59.2/3.5 | 55.8 |
| **34B** | $10 \times 12 \times 12$ | 58.5/3.5 | 50.8 |
|  | $5 \times 24 \times 24$ | 58.3/3.5 | 54.4 |
|  | $5 \times 12 \times 24$ | 57.2/3.4 | 51.2 |
|  | $5 \times 12 \times 12$ | 55.6/3.4 | 47.8 |

(a) Results on SF-LLaVA-7B models.  (b) Results on SF-LLaVA-34B models.

Table 5: *Effect of applying different pooling strategies on the Slow pathway of SF-LLaVA*, where symbol "–" means the model inference gets out-of-memory on 80GB GPUs under this setting.

| Model Size | Output #tokens | ANet-QA | EgoSchema |
|---|---|---|---|
|  | $50 \times 8 \times 8$ | 54.9/3.3 | 48.6 |
|  | $50 \times 4 \times 8$ | 55.0/3.3 | 46.8 |
|  | $50 \times 6 \times 6$ | 55.2/3.3 | 46.2 |
| **7B** | $50 \times 3 \times 6$ | 54.9/3.3 | 46.6 |
|  | $50 \times 4 \times 4$ | 55.5/3.3 | 47.2 |
|  | $50 \times 2 \times 4$ | 54.7/3.3 | 47.4 |
|  | $50 \times 1 \times 1$ | 54.4/3.3 | 47.1 |

| Model Size | Output #tokens | ANet-QA | EgoSchema |
|---|---|---|---|
|  | $50 \times 8 \times 8$ | - | - |
|  | $50 \times 4 \times 8$ | - | - |
|  | $50 \times 6 \times 6$ | - | - |
| **34B** | $50 \times 3 \times 6$ | 58.5/3.5 | 55.3 |
|  | $50 \times 4 \times 4$ | 59.2/3.5 | 55.8 |
|  | $50 \times 2 \times 4$ | 58.6/3.5 | 55.2 |
|  | $50 \times 1 \times 1$ | 58.9/3.5 | 56.0 |

(a) Results on SF-LLaVA-7B models.  (b) Results on SF-LLaVA-34B models.

Table 6: *Effect of applying different pooling strategies on the Fast pathway of SF-LLaVA*, where symbol "–" means the model inference gets out-of-memory on 80GB GPUs under this setting.

| Model Size | Task Instruction Prompt | Input Data Prompt | Structured Answer Prompt | ANet-QA | EgoSchema |
|---|---|---|---|---|---|
|  | ✔ | ✔ | ✔ | 54.9/3.3 | **47.2** |
| **7B** | ✗ | ✔ | ✔ | **55.5/3.4** | 43.0 |
|  | ✔ | ✗ | ✔ | 52.6/3.2 | 44.8 |
|  | ✔ | ✔ | ✗ | 52.7/3.4 | 44.4 |
|  | ✔ | ✔ | ✔ | 58.4/3.5 | **55.8** |
| **34B** | ✗ | ✔ | ✔ | **59.2/3.5** | 52.2 |
|  | ✔ | ✗ | ✔ | 56.4/3.3 | 55.4 |
|  | ✔ | ✔ | ✗ | 58.5/3.5 | 55.6 |

Table 7: *Results of using different prompt designs for SF-LLaVA.*

**Pooling impact on Fast pathway.** We keep $\mathbf{F}_v^{\text{slow}} \in \mathbb{R}^{10 \times 12 \times 24}$ and $N^{\text{fast}} = 50$, while reducing $\mathbf{F}_v^{\text{fast}}$ to $\{8 \times 8, 4 \times 8, 6 \times 6, 3 \times 6, 4 \times 4, 2 \times 4, 1 \times 1\}$ tokens by pooling. Note that $1 \times 1$ is an extreme case that loses all spatial information and can only contribute to temporal modeling. Table 6 shows that keeping more tokens, such as $8 \times 8$, generally gets better results on average, but the performance gap is not obvious. Considering that ($i$) SF-LLaVA-34B cannot afford more than $50 \times 3 \times 6$ output tokens due to out-of-memory and ($ii$) 50 frames are able to cover most videos in existing benchmarks, we use $4 \times 4$ as our default setting to trade-off between spatial and temporal information. However, if we extend SF-LLaVA for long-form video understanding (*e.g.,* over 30 minutes), using $1 \times 1$ in the Fast pathway could be a better choice to cover more input frames.

**Number of frames in Fast pathway.** We evaluate the effect of using different $N^{\text{fast}}$. Similar to the above experiments, we keep the Slow pathway as its default in Sec. 4.2, and test to increase $N^{\text{fast}}$ from 10 to 50 frames. Note that we chose 50 frames as our maximum because this reaches the GPU memory limit for the 34B model. The results in Fig. 4 show that using more frames in Fast pathway improves the performance on both ActivityNet-QA and EgoSchema datasets (*e.g.,* using 50 frames outperforms 10 frames by 1.7% and 2.8% on EgoSchema on 7B and 34B models, respectively). Thus, by default, SF-LLaVA uses $N^{\text{fast}} = 50$ in the Fast pathway.

### 4.5 DESIGN CHOICES OF PROMPT

SF-LLaVA is built upon a pre-trained Image LLM for VideoQA without further fine-tuning. Here we evaluate if we should design new prompts for SF-LLaVA to better understand the video inputs and tasks. We decompose the prompt into three main parts and respectively explore their best designs.

**Task instruction prompt** clarifies the goal of the target task. In particular, we use "Answer the question precisely based on the input" for the open-ended task, and "Select the best option to answer the question" for the multiple choice task. Table 7 (row 1 and 2) shows that using the task instruction prompt can improve the performance on EgoSchema (47.2% *v.s.* 43.0% and 55.8% *v.s.* 52.2% on 7B and 34B) but is not helpful for ActivityNet-QA (54.9% *v.s.* 55.5% and 58.4% *v.s.* 59.2% on 7B and 34B). Thus, SF-LLaVA uses the task instruction prompt *only* for Multiple Choice VideoQA.

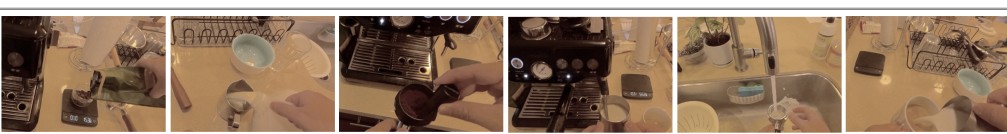

**Question 1:** What is happening in this video?

**SlowFast-LLaVA-7B:** The video depicts a person preparing a cup of coffee, which is a common activity in the morning or during breaks. The coffee machine is a modern, sleek design with a digital display. The kitchen is well-equipped with various appliances and utensils. There's a sink with a faucet, a dishwasher, and a refrigerator. On the countertop, there are several items including a bowl, a spoon, and a cup. The overall content of the video is a snapshot of everyday life, capturing the simple yet essential act of making coffee in a home setting.

**Question 2:** When does the person wash the portafilter?

**SlowFast-LLaVA-7B:** The person washes the portafilter at the end of the video, after preparing the coffee.

**Question 3:** Does this person open the refrigerator?

**SlowFast-LLaVA-7B:** No, the person in the video does not open the refrigerator.

Figure 5: *Qualitative results of SF-LLaVA-7B.* Our model can correctly understand the video content and capture important details. On the other hand, error cases show that SF-LLaVA lacks the capability of fine-grained temporal grounding and may miss certain moments due to uniform sampling.

**Input data prompt** describes the structure of the inputs, such as the image grid in IG-VLM (Kim et al., 2024) and image sequence in PLLaVA (Xu et al., 2024). For all tasks, SF-LLaVA uses the same prompt "The input consists of a sequence of key frames from a video". Table 7 (row 1 and 3) shows that using input data prompt obtains better results on both ActivityNet-QA (54.9% *v.s.* 52.6% and 58.4% *v.s.* 56.4%) and EgoSchema (47.2% *v.s.* 44.8% and 55.8% *v.s.* 55.4%). This demonstrates the importance of offering input data details to better understand the structure of visual tokens, which we think is especially important to training-free methods.

**Structured answer prompt** guides Video LLMs to generate answers in a more desirable format. This is especially important to the Multiple Choice VideoQA task, since it makes the answer easier to be parsed and improves the performance out of the box (Li et al., 2023d). We follow MVBench (Li et al., 2023d) to use "Best Option:(" as the answer prompt for Multiple Choice VideoQA and follow Image Grid (Kim et al., 2024) to use "In this video," to guide the Open-Ended VideoQA tasks. Table 7 (row 1 and 4) shows that using structured answer prompts improves results by 2.2% on ActivityNet-QA and 2.8% on EgoSchema with the 7B LLM.

## 4.6 ERROR ANALYSIS

First, although SF-LLaVA can understand the relative sequence of different video moments, it still lacks the capability to detect their precise start and end time, such as the Question 2 in Fig. 5. This is because SF-LLaVA is never trained on any video datasets and relevant tasks. Fine-tuning SF-LLaVA on fine-grained time-related video datasets could be a promising direction to gain this capability, and incorporating multimodal inputs (*e.g.,* the timestamp, subtitle, and audio of each frame) can further improve the performance. Second, as observed in many other methods, SF-LLaVA possibly misses some key frames due to its uniform frame sampling. Question 3 in Fig. 5 is an example, in which the frames showing a quick moment of "opening refrigerator" are unintentionally missed (thus not shown in the figure). Drastically sampling more frames (*e.g.,* in 5 FPS) can mitigate this issue but is limited by an LLM's context window. This suggests we may explore dynamic sampling strategies to ensure a more comprehensive sampling of important video segments.

## 5 CONCLUSION

We present SF-LLaVA, a training-free Video LLM that is built upon LLaVA-NeXT and requires no additional fine-tuning to work effectively for various video tasks. Especially, we propose a SlowFast design that uses two-stream inputs for Video LLMs. It aggregates frame features as an effective video representation that can capture both detailed spatial semantics and long-range temporal context. Our experiments on a diverse set of 8 video benchmarks demonstrate the effectiveness of SF-LLaVA, where it outperforms existing training-free methods. On some benchmarks, SF-LLaVA achieves on-par or even better results than state-of-the-art SFT Video LLMs that have been extensively fine-tuned on large-scale video data. We hope SF-LLaVA can serve as a simple but strong baseline in the whole picture of Video LLMs, and our ablation on its design choices can provide valuable insights for future research on modeling video representations for Multimodal LLMs.

## REPRODUCIBILITY STATEMENT

We have made every effort to include as many implementation details as possible in Sec. 4.2. We commit to releasing the source code upon publication.

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
