# OpenReview forum: "SlowFast-LLaVA: A strong training-free baseline for video large language models"
_ICLR.cc/2025/Conference — Submitted to ICLR 2025_

### Official Review · Reviewer_zed3 · 2024-10-26

**Soundness:** 3
**Presentation:** 2
**Contribution:** 3
**Rating:** 6
**Confidence:** 4

**Summary:**

Inspired by the idea of two stream networks in action recognition, this paper introduces a really simple sampling scheme of the frames and spatial pooling for training free evaluation of image LLM on videos. They show consistent improvements over the baseline on many tasks.

**Strengths:**

The idea is extremely simple (which is a good thing) and battle tested in action recognition.

The evaluations and ablations seem comprehensive and fair.

**Weaknesses:**

The paper suffers from excessive repetition. Given the simplicity of the core idea, it seems unnecessarily stretched to fill 10 pages and could be more concise. Additionally, table captions, such as in Table 1, lack added value; they simply reiterate the table content without providing further insights. Similarly, the experiment descriptions merely repeat the accuracy values from the tables, without offering additional intuition or interpretation.

Furthermore, the prompt aspect could be explored in greater depth, as it represents an additional axis to consider in zero-shot adaptation of vision-based large language models (LLMs) for videos.

**Questions:**

It seems that the prompt aspect in zero-shot adaptation could add substantial value if further explored. Have you considered additional prompt engineering or variations in the prompts to evaluate their impact on the model’s performance?

---

> ### Author Response · Authors · 2024-11-20
> **Response to Reviewer zed3**
>
> We thank the reviewer for their insightful comments. We address your comments in the following.
>
> ---
>
> **Q1:**
> The paper suffers from excessive repetition. Given the simplicity of the core idea, it seems unnecessarily stretched to fill 10 pages and could be more concise. Additionally, table captions, such as in Table 1, lack added value; they simply reiterate the table content without providing further insights. Similarly, the experiment descriptions merely repeat the accuracy values from the tables, without offering additional intuition or interpretation.
> Furthermore, the prompt aspect could be explored in greater depth, as it represents an additional axis to consider in zero-shot adaptation of vision-based large language models (LLMs) for videos.
>
> **A1:**
> Thank you for these suggestions. We will improve our paper writing in the revision. Specifically, we will
>
> - prune the "repeating of accuracy values" from the tables and experiments
> - summarize experimental results and give more result interpretations
> - report more results and insights about our better zero-shot ability, as suggested by **Reviewer Hn2a**
> - report more results and discussions about our advantage on long-form video understanding, as suggested by **Reviewer Hn2a** and **yZMr**
> - discuss the possibility to extend our SlowFast design on other Image LLMs and report results, as suggested by **Reviewer jACj** and **yZMr**
>
> Please let us know if you have other specific suggestions and we will take care of them together in the revision.
>
> ---
>
> **Q2:**
> It seems that the prompt aspect in zero-shot adaptation could add substantial value if further explored. Have you considered additional prompt engineering or variations in the prompts to evaluate their impact on the model’s performance?
>
> **A2:**
> Thank you for this suggestion. Our prompt exploration is intended to give helpful insights for future work on how to systematically design their own prompts in zero-shot domain adaptation. We evaluated the importance of prompts from task, input, and answer perspectives, and showed their impacts on various VideoQA tasks. We agree that additional prompt engineering possibly further improves the performance of our model and inspires the design of future work. We will discuss more about our prompt experiments in the revision.

---

> > ### Comment · Reviewer_zed3 · 2024-11-20
> >
> > Thanks for your answers, I'm much happier now with all the new experiments that you conducted (to answer other reviewers) and will raise my score.

---

> > > ### Author Response · Authors · 2024-11-20
> > > **Response to Reviewer zed3**
> > >
> > > Thank you for raising your score!

---

### Official Review · Reviewer_yZMr · 2024-10-31

**Soundness:** 2
**Presentation:** 3
**Contribution:** 3
**Rating:** 6
**Confidence:** 4

**Summary:**

This paper presents SF-LLaVA, a training-free video LLM that requires no additional fine-tuning to work for various video tasks.  Technically, the authors propose a SlowFast design of inputs for Video LLMs to capture both detailed spatial semantics and long-range temporal context, which is a simple and effective approach. The experiments on 3 video tasks with 8 benchmarks demonstrate the effectiveness and superiority of SF-LLaVA compared to training-free and SFT methods. Comprehensive ablation studies provide valuable insights for future research.

**Strengths:**

+ The motivation for enabling LLMs to understand video inputs through a training-free approach is reasonable.

+ The paper is easy to read and well-structured.

+ Experiments on a diverse set of 8 video benchmarks demonstrate the effectiveness of SF-LLaVA.

**Weaknesses:**

- Experiments were conducted only on LLaVA-NeXT (despite including 7B and 34B models), with no additional results from other open-source LLMs provided.


- The quantitative results in Figure 5 lack the results without the SlowFast design.

**Questions:**

-  Can the proposed method work for particularly long video understanding, such as LVBench[1]?

-  The paper indicates that it lacks the capability to detect the precise start and end times. Can fine-grained frame-level textual descriptions be introduced to enhance the spatio-temporal understanding of videos?

---

> ### Author Response · Authors · 2024-11-20
> **Response to Reviewer yZMr**
>
> We thank the reviewer for their insightful comments. We address your comments in the following.
>
> ---
>
> **Q1:**
> Experiments were conducted only on LLaVA-NeXT (despite including 7B and 34B models), with no additional results from other open-source LLMs provided.
>
> **A1:**
> We focus on LLaVA-style Image LLMs since they are among the most popular model architectures. But we agree that validating our SlowFast idea on other Image LLMs would be beneficial. As suggested, we integrate our SlowFast inputs to Phi-3 Vision (named SlowFast-Phi-3), which is the state-of-the-art mobile-size (4.2B) Image LLM with pixel shuffle structure, and evaluated it on various benchmarks. We directly use the design choices and hyperparameters of SF-LLaVA-7B without ablation studies. To better understand the results of this model, we also implement IG-VLM for Phi-3 Vision as baseline (named IG-Phi-3), which is the state-of-the-art method of adopting Image LLMs for video tasks. The table below shows that SlowFast-Phi-3 achieves convincing results on all benchmarks and outperforms Image Grid by a clear margin.
>
> | Model    | VideoMME | MVBench | EgoSchema | ActivitiyNet-QA | IntentQA
> | -------- | :--------: | :-------: | :--------: | :-------: | :--------:
> | IG-Phi-3  | 44.5%   | 43.5% | 37.6%   | 53.1%  | 60.6%
> | SlowFast-Phi-3 | **47.4%** | **45.0%** | **39.2%** | **53.3%** | **61.4%**
>
> ---
>
> **Q2:**
> The quantitative results in Figure 5 lack the results without the SlowFast design.
>
> **A2:**
> Thanks for this suggestion. Here we report the answers of LLaVA-7B which simply takes as inputs 12 video frames in a sequence. This baseline has similar number of tokens as SF-LLaVA-7B in paper's Figure 5. By comparing the results of these two models, we observe that *i)* SF-LLaVA-7B gives more details than LLaVA-7B in Question 1, *ii)* LLaVA-7B fails to understand the relative order of different events without the SlowFast design in Question 2, and *iii)* SF-LLaVA-7B and LLaVA-7B both miss the "opening refrigerator" event due to the uniform frame sampling in Question 3. We will include these results in our revision.
>
> - **Question 1:** What is happening in this video?
>
> - **LLaVA-7B:** In this video, a person is preparing a coffee using a coffee machine. The scene shows the coffee machine being filled with coffee grounds, the coffee being poured into a cup, and the coffee being stirred. The setting seems to be a kitchen or a similar environment with various kitchen items visible in the background. The overall content of the video is focused on the preparation of a hot beverage, likely coffee, in a domestic setting.
>
> - **Question 2:** When does the person wash the portafilter?
>
> - **LLaVA-7B:** The exact time when the portafilter is being washed cannot be determined from the video, but it is clear that the person is engaged in this task during the coffee-making process.
>
> - **Question 3:** Does this person open the refrigerator?
> - **LLaVA-7B:** This person is not shown opening the refrigerator in this video.
>
> ---
>
> **Q3:**
> Can the proposed method work for particularly long video understanding, such as LVBench[1]?
>
> **A3:**
> As suggested, we evaluated SF-LLaVA on LongVideoBench. We also evaluated IG-VLM as our training-free baseline by using their official codebase. The table below shows that SF-LLaVA outperforms IG-VLM by **+3.5% on LongVideoBench** and achieves even better performance than PLLaVA (fine-tuned on VideoChat2 with ~900K training samples), using both 7B and 34B LLMs. We will report these numbers in our revision and give more discussions.
>
> | Model    | LongVideoBench (Validation Set)
> | -------- | :--------:
> | IG-VLM-7B  | 38.1%
> | PLLaVA-7B  | 40.2%
> | SF-LLaVA-7B | **41.3%**
> | IG-VLM-34B  | 49.8%
> | PLLaVA-34B  | 53.2%
> | SF-LLaVA-34B | **53.9%**
>
> **Table 1:** The results of IG-VLM are reproduced by using their official codebase and the results of PLLaVA are copied from the official LongVideoBench leaderboard.
>
> ---
>
> **Q4:**
> The paper indicates that it lacks the capability to detect the precise start and end times. Can fine-grained frame-level textual descriptions be introduced to enhance the spatio-temporal understanding of videos?
>
> **A4:**
> Thanks for this great suggestions. We believe using frame-level description can improve the spatial-temporal understanding of a video, which is already suggested by many existing methods such as LLoVi and VideoTree. These information will help the Video LLM to better understand what is happening in each frame and precisely detect a target event. On the other hand, we also believe incorporating the timestep of each input video frame is important for detecting an event's start and end times, especially when we want the outputs to be in seconds, not the frame number. We will give more discussion about this topic in our final revision.

---

> > ### Author Response · Authors · 2024-11-21
> >
> > Dear Reviewer yZMr,
> >
> > Thanks again for your valuable feedback! Feel free to let use know if you have more questions. We are looking forward to your feedback!

---

### Official Review · Reviewer_jACj · 2024-11-02

**Soundness:** 3
**Presentation:** 3
**Contribution:** 2
**Rating:** 5
**Confidence:** 4

**Summary:**

The paper proposes SlowFast-LLaVA (SF-LLaVA), a training-free video large language model (LLM) that can jointly capture detailed spatial semantics and long-range temporal context without exceeding the token budget of commonly used LLMs. This is achieved by using a two-stream SlowFast design of inputs, where the Slow pathway extracts features at a low frame rate while keeping spatial detail, and the Fast pathway operates on a high frame rate but uses a larger spatial pooling stride to focus on motion cues. Experimental results show that SF-LLaVA outperforms existing training-free methods on a wide range of video tasks, and achieves comparable or even better performance compared to state-of-the-art video LLMs that are fine-tuned on video datasets.

**Strengths:**

1. the proposed method to build video-llm is simple, novel, and the slowfast design to aggregate spatial and temportal information is reasonable.
2.the paper is in good writing for understanding, and ablation experiments to provide detailed results for the influential factors of the provided method.

**Weaknesses:**

1. video-mllms could not only do recgnition and captioning of a video, but also exploite the knowledge of LLM to do further recognition reseasoning. In order to comprehensively evaluate the proposed training-free method to build a video-mllm.

a. I suggest authors provide evaluations not only on captioning benchmarks, such as MSVD, MSR-VTT and so on, but also on benchmarks specifically for video-mllm, such as video-mme, mvbench, videovista, MLVU and so on, which could provide more comprehensive video-llm performance comparation, the evaluation resutls could provide important new insights about the model's performance.

2. the two-stream method is only applied on llava-next image-llm, if authors could give proof that this idea could be generalized to other image-llm models, the two-stream token aggregation design is more persuasive and thought-provoking for futher video-llm research. Here, I list some recent  image-llm models with structure different from llava, blip2(with q-former structure), internvl(with pixel shuffle structure), Ovis(Ovis: Structural Embedding Alignment for
 Multimodal Large Language Model, with visual embedding table), Aria（An Open Multimodal Native Mixture-of-Experts Model， with MoE structure）and suggest authors give test or theoretical analysis on these models.

3.Althogh training-free is a good feature for the method, a training-based model could gain a more powerful performance, just as  LLaVA-NeXT-Video-DPO is better than SF-LLaVA(Both built on LLaVA-NeXT-Image ).

a. I suggest authors discuss specific use cases or scenarios where a training-free approach might be preferable despite potentially lower performance.

b. I also suggest authors elaborate on potential trade-offs between training-free and fine-tuned approaches in terms of factors like deployment flexibility, computational requirements, or ability to handle novel domains.

**Questions:**

questions are listed above

---

> ### Author Response · Authors · 2024-11-20
> **Response to Reviewer jACj**
>
> We thank the reviewer for their insightful comments. We address your comments in the following.
>
> ---
>
> **Q1:**
> I suggest authors provide evaluations not only on captioning benchmarks, such as MSVD, MSR-VTT and so on, but also on benchmarks specifically for video-mllm, such as video-mme, mvbench, videovista, MLVU and so on, which could provide more comprehensive video-llm performance comparation, the evaluation resutls could provide important new insights about the model's performance.
>
> **A1:**
> As suggested, we evaluate SF-LLaVA on VideoMME and MVBench as shown below. We observe that SF-LLaVA-34B achieves very convincing results compared with other training-based models in the leaderboard, such as InternVL-Chat-V1.5. To compare with training-free baselines, we evaluated IG-VLM by using their official codebase. We find that SlowFast-LLaVA outperforms this strong baseline on these benchmarks by a clear margin.
>
> | Model | VideoMME (Average) | MVBench (Average)
> | --- | :---: | :---:
> | IG-VLM-7B  | 39.8%   | 41.5%
> | SF-LLaVA-7B | **41.2%**   | **44.1%**
> | IG-VLM-34B  | 50.9%   | 48.1%
> | SF-LLaVA-34B | **52.9%**  | **49.8%**
>
> Due to the limited time, we won't benchmark VideoVISTA and MLVU in this rebuttal. We will cite them in the paper revision and use them in our future work.
>
> ---
>
> **Q2:**
> the two-stream method is only applied on llava-next image-llm, if authors could give proof that this idea could be generalized to other image-llm models, the two-stream token aggregation design is more persuasive and thought-provoking for futher video-llm research.
>
> **A2:**
> We focus on LLaVA-style Image LLMs since they are among the most popular model architectures. But we agree that validating our SlowFast idea on other Image LLMs would be beneficial. As suggested, we integrate our SlowFast inputs to Phi-3 Vision (named SlowFast-Phi-3), which is the state-of-the-art mobile-size (4.2B) Image LLM with **pixel shuffle structure**, and evaluated it on various benchmarks. We directly use the design choices and hyperparameters of SF-LLaVA-7B without ablation studies. To better understand the results of this model, we also implement IG-VLM for Phi-3 Vision as baseline (named IG-Phi-3), which is the state-of-the-art method of adopting Image LLMs for video tasks. The table below shows that SlowFast-Phi-3 achieves convincing results on all benchmarks and outperforms Image Grid by a clear margin.
>
> | Model    | VideoMME | MVBench | EgoSchema | ActivitiyNet-QA | IntentQA
> | --- | :---: | :---: | :---: | :---: | :---:
> | IG-Phi-3  | 44.5%   | 43.5% | 37.6%   | 53.1%  | 60.6%
> | SlowFast-Phi-3 | **47.4%** | **45.0%** | **39.2%** | **53.3%** | **61.4%**
>
> Due to the limited time, we will leave the implementation of SlowFast into other suggested Image LLMs in the revision.
>
> ---
>
> **Q3:**
> Althogh training-free is a good feature for the method, a training-based model could gain a more powerful performance, just as LLaVA-NeXT-Video-DPO is better than SF-LLaVA(Both built on LLaVA-NeXT-Image ).
>
> - I suggest authors discuss specific use cases or scenarios where a training-free approach might be preferable despite potentially lower performance.
> - I also suggest authors elaborate on potential trade-offs between training-free and fine-tuned approaches in terms of factors like deployment flexibility, computational requirements, or ability to handle novel domains.
>
> **A3:**
> Thanks for this great question. *First,* training/fine-tuning Video LLMs on large-scale video datasets requires high computation cost. For example, fine-tuning PLLaVA-7B with lightweight parameters using LoRA requires about 35 hours on 8 A100-80GB GPUs. This resource cannot be afforded by many small company and university labs. On the other hand, it also involves video data collection from a legally approved source, human/machine annotation, related training infra support and quality verification. Thus we believe developing training-free Video LLMs can be beneficial for our community. *Second,* fine-tuning Image LLMs on video will be unavoidable to hurt their results on image tasks. Video-LLaVA tries to solve this by using separate image and video encoders, which increases the model size in the deployment. Another option is to train a unified model, such as LLaVA-OneVision, but this model will possibly require an even larger mixed image and video datasets (*e.g.,* 5M+ samples), multi-round stages, and even more computation resources. Since the training-free methods are never fine-tuned on video data, we can simply deploy a single model on server/device working for different application scenarios. *Third,* as mentioned by **Reviewer Hn2a**, since existing Video LLMs are usually fine-tuned on in-domain video datasets, they likely have poor generalization capabilities on other domains. Training-free methods, such as SF-LLaVA and IG-VLM, show great zero-shot video question-answering capabilities on various benchmarks.

---

> > ### Author Response · Authors · 2024-11-21
> >
> > Dear Reviewer jACj,
> >
> > Thank you for your helpful comments! Please let us know if your concerns have been fully addressed. We are looking forward to your feedback!

---

> > > ### Author Response · Authors · 2024-11-29
> > >
> > > Dear Reviewer jACj,
> > >
> > > Thanks again for your valuable feedback! As we are close to the rebuttal deadline, we would like to double check with you if our response resolved your major concern and if you are willing to raise your score? Thank you!
> > >
> > > Authors of Submission610

---

> > > > ### Comment · Reviewer_jACj · 2024-11-30
> > > >
> > > > Thank you for your diligent answers, I still have concerns which make me retain my score.
> > > > 1. On insight part. Although, SF-LLaVAs have shown superior performance to IG-VLM, they are quite inferior to top-level models with same model size in the MVBench and Video-MME leaderboard. The value of this paper should be on the insight from the slow-fast feature aggregation. Interesting experimental results can be seen from the paper, but there is a lack of in-depth analysis of the results.  I am not convinced that this method could really model the motion information of a video.
> > > > 2. On application part. Zero-shot video question-answering capabilities is not solidly verified.
> > > > 3. On generalization part. I am not conviced this aggregation method could be generalized to other various image-llm models with different structures.
> > > >
> > > > Overall, my main concern is that the paper does not provide sufficiently in-depth insight.

---

> > > > > ### Author Response · Authors · 2024-11-30
> > > > > **Response to Reviewer jACj**
> > > > >
> > > > > Thank you for reply. We respectfully disagree with your comments and would like to further clarify as follows.
> > > > >
> > > > > **Q4:** On insight part. Although, SF-LLaVAs have shown superior performance to IG-VLM, they are quite inferior to top-level models with same model size in the MVBench and Video-MME leaderboard. The value of this paper should be on the insight from the slow-fast feature aggregation. Interesting experimental results can be seen from the paper, but there is a lack of in-depth analysis of the results. I am not convinced that this method could really model the motion information of a video.
> > > > >
> > > > > **A4:** *First,* we explained in **your previous Q3** about the advantages of developing training-free Video LLMs, such as their very low cost in both computation and data, their better generalization ability, and their better practice in deploying a unified image and video LLMs. *Second,* as a training-free method, SF-LLaVA achieves on-par or even better results than many training-based methods in the MVBench and Video-MME leaderboard, such as VideoChat2-Mistral and ShareGPT4Video, which we believe is already a great achievement in performance. We kindly remind that MVBench and Video-MME are just two of the various VideoQA benchmarks, and SF-LLaVA also achieves consistently great performance on other standard benchmarks. *Third,* in our response to **Reviewer Hn2a Q5**, we tested to shuffle the order of video tokens and found the performance of SF-LLaVA drops on ActivityNet-QA and EgoSchema, showing that motion information is important for SF-LLaVA to achieve the best performance on these benchmarks. More importantly, in that response, we provided the insights that, the performance gap on ActivityNet-QA is much smaller than EgoSchema (1.4% v.s. 3.6%). We argue this is because many questions in ActivityNet-QA, such as "indoor or outdoor" and "object counting", mostly rely on spatial recognition, and using tokens as a bag of words could be sufficient to answer them. EgoSchema is proposed for complex long-form temporal reasoning, thus shuffling the order of tokens has more negative impacts on its results. We also provided a future work that the performance gap of using normal and shuffled order of tokens can be used as a metric to evaluate the quality of the video benchmarks.
> > > > >
> > > > > ---
> > > > >
> > > > > **Q5:** On application part. Zero-shot video question-answering capabilities is not solidly verified.
> > > > >
> > > > > **A5:** *First,* we kindly remind that **all of our experiments are under the zero-shot setting**, since SF-LLaVA is never trained on any video dataset. On the other hand, other training-based baselines can use some in-domain raw videos or training datasets, such as both PLLaVA and VILA1.5 are trained on raw videos of ActivityNet. *Second,* as suggested by **Reviewer Hn2a Q7*, we validate if SF-LLaVA achieves better performance under the zero-shot setting. The following observations support our hypothesis.
> > > > >
> > > > > - PLLaVA-7B uses VideoChat2, which includes raw videos from ActivityNet and TGIF. Thus, its performance gap compared to SF-LLaVA-7B on these two benchmarks are relatively small (< 1%). However, on other open-ended benchmarks, such as MSRVTT, SF-LLaVA-7B outperforms PLLaVA-7B largely (+3.8%).
> > > > > - VILA1.5 is trained on ShareGPTVideo (including raw videos from ActivityNet), thus, its performance gap compared to SF-LLaVA is relatively small on ActivityNet-QA (about 1%). However, on TGIF-QA, where VILA1.5 was not trained on, SF-LLaVA significantly outperforms it by over +20%.
> > > > >
> > > > > ---
> > > > >
> > > > > **Q6:** On generalization part. I am not conviced this aggregation method could be generalized to other various image-llm models with different structures.
> > > > >
> > > > > **A6:** *First,* as indicated by **our title**, we intend to focus on LLaVA-style structure. This is because LLaVA uses the simplest and most general model structure and is among the most popular Image LLMs. *Second*, in **our previous answer to your Q2**, we have followed your suggestion and integrated our SlowFast design into Phi-3 Vision which uses a different structure compared to LLaVA. By comparing with IG-VLM, we showed that SlowFast-Phi-3 achieves convincing results, which has verified that our SlowFast design can be applied to other Image LLMs. Please understand that, due to the limited time of rebuttal, we cannot provide more results on other Image LLMs.
> > > > >
> > > > > ---
> > > > >
> > > > > We will add above discussions in our revision and give more insights for future work. Could you please let us know if the above clarification resolves your concerns? We are looking forward to further discussion. Thank you!

---

### Official Review · Reviewer_Hn2a · 2024-11-04

**Soundness:** 3
**Presentation:** 3
**Contribution:** 2
**Rating:** 5
**Confidence:** 4

**Summary:**

This paper proposes a training free method that converts an image MLLM into a video MLLM. This was achieved by simply concatenating features or frames in previous works. This paper proposes to concatenate features in a slow fast manner that combines a few high resolution feature maps with a large number of low resolution feature maps, which enabled LLM to see both local details and temporal context. The method SF-LLAVA achieves improved results on video MLLM benchmarks.

**Strengths:**

1. The proposed method is simple, intuitive, and efficient. It involves no training but obtains a video MLLM for free.
2. Ablations are performed extensively on multiple datasets and benchmarks, focusing on the optimal selection of the slow and fast resolutions.

**Weaknesses:**

1. Limited novelty compared with existing methods such as IG-VLM, or PLLaVA. It has been shown that simply concatenating frames (or frame features) gives a good MLLM without or with only a little training. On top of these existing works, it is straightforward to see or expect that a cleverer selection of feature such as adopting the SlowFast idea can work a little bit better.
2. Limited performance gain on top of the baselines such as IG-VLM, as shown in Table 2 (e.g. +1.1% on NextQA), as well as in the ablation, line 413, where adding the fast path improves marginally on top of the 12 frame slow only model.
3. It is less well justified to keep “training free” as a requirement. It is true that training free is a nice property but existing training-based methods such as pllava and VILA involve only a lightweight fine-tuning stage that initializes mostly from image (LLaVA) model and learns only a projector or arguably simply concept selection/merging layers.
4. Limited comparison with strong training based methods, such as VILA, and more comprehensive benchmarks such as MVBench or VideoMME. The benchmarks that are currently studied are either spatial biased (i.e. a 1-frame model can do quite well) or temporal-biased (EgoSchema that is based on high level semantics), rather than those that require both spatial and temporal reasoning.
5. It may be interesting to see the difference in the order of input tokens. The current model looks like a bag of words model with the LLM doing most of the heavy lifting. Does the order of the tokens really matter? Does the model really learn motion? Do the datasets require/measure motion? This may give more insights to the community, if the goal is not to push the best results on video benchmarks.

**Questions:**

1. Since SF-LLAVA does not require training, is it possible to compute features offline to save some memory and thus use a lot more tokens than those methods that compute vision features during inference? In this case, is it possible to show that SF-LLAVA can work as a general add-on to any model that improves inference for any model? e.g. trained on 4 frames but can infer on many more frames with SF-Model?
2. Does SF-LLaVA achieve good results on zero-shot generalization cases, compared with video trained methods such as pllava and vila? e.g. these methods are trained on specific datasets and therefore work well on those datasets that are similar to the training distribution, while SF-LLAVA works out of box and better on some datasets that are more different than the training sets of these existing papers.
3. Following the discussion on weakness #3, is it possible to apply SF-LLaVA on larger scale models where video training is less feasible such as a 70B/90B llama3 model? This may be a real advantage of training-free vs. training-based methods.

---

> ### Author Response · Authors · 2024-11-20
> **Response to Reviewer Hn2a (1/3)**
>
> We thank the reviewer for their insightful comments. We address your comments in the following.
>
> ---
>
> **Q1:**
> Limited novelty compared with existing methods such as IG-VLM, or PLLaVA. It has been shown that simply concatenating frames (or frame features) gives a good MLLM without or with only a little training. On top of these existing works, it is straightforward to see or expect that a cleverer selection of feature such as adopting the SlowFast idea can work a little bit better.
>
> **A1:**
> *First,* we understand your concern. However, the significance of a research paper extends beyond the introduction of novel training methods. We believe that the contributions of our work are substantial, and the findings presented in the paper are insightful. Specifically, we present **a simple yet effective** SlowFast input design tailored for Video LLMs, which achieves surprisingly competitive performance. This approach can serve as a strong baseline for all future research in the field.
>
> *Second,* compared with IG-VLM and PLLaVA, SF-LLaVA uses a totally different input structure. *i)* IG-VLM assembles multiple video frames into an image grid and inputs it as a high-resolution image into the MLLM. Limited by the image grid size, IG-VLM is only effective to consume 6 frames (their performance decreases when using more frames). *ii)* PLLaVA simply concatenates the pooled video features into a sequence without proper temporal modeling designs, thus cannot further improve the performance using more than 16 frames. *iii)* Our two-stream SlowFast inputs can effectively capture long-range context while preserving detailed spatial semantics. As a result, SF-LLaVA takes over 50 frames as inputs and achieves convincing results on various video tasks.
>
> *Third,* we respectfully disagree that SF-LLaVA works *only a little bit better* than simply concatenating frames. *i)* SF-LLaVA-7B outperforms IG-VLM-7B on **all 8 video benchmarks**, especially by **+11.4% on EgoSchema** and **+5.7% on TGIF-QA**. *ii)* As a training-free method, SF-LLaVA achieves on-par performance with PLLaVA, which is fine-tuned on VideoChat2 with 15 video datasets and ~900K training samples. Please also note that VideoChat2 contains raw videos from ActivityNet and TGIF, thus these results are not out-of-domain as SF-LLaVA. *iii)* To strengthen our superiority on long-form video understanding, we evaluate SF-LLaVA on LongVideoBench that contains videos up to an hour long. The table below shows that SF-LLaVA outperforms IG-VLM by **+3.2% and +4.1% on LongVideoBench** using 7B and 34B LLMs, and achieves even better performance than PLLaVA.
>
> | Model | EgoSchema | LongVideoBench (Validation Set)
> | -------- | :--------: | :--------------:
> | IG-VLM-7B | 35.8% | 38.1%
> | PLLaVA-7B | - | 40.2%
> | SF-LLaVA-7B | **47.2%** | **41.3%**
> | IG-VLM-34B | 53.6% | 49.8%
> | PLLaVA-34B | - | 53.2%
> | SF-LLaVA-34B | **55.8%** | **53.9%**
>
> **Table 1:** The results of IG-VLM are reproduced by using their offical codebase and the results of PLLaVA are copied from the official LongVideoBench leaderboard.
>
> ---
>
> **Q2:**
> Limited performance gain on top of the baselines such as IG-VLM, as shown in Table 2 (e.g. +1.1% on NextQA), as well as in the ablation, line 413, where adding the fast path improves marginally on top of the 12 frame slow only model.
>
> **A2:**
> we respectfully disagree our performance gain on top of IG-VLM is marginal. *First,* SF-LLaVA consistently outperforms IG-VLM on **all 8 video benchmarks**, using both 7B and 34B LLMs. *Second,* NExTQA mostly contains short videos (44s on average), which cannot fully show SF-LLaVA's superior capability on modeling long-range context. We kindly remind that SF-LLaVA-7B outperforms IG-VLM-7B by **+11.4% on EgoSchema**, which contains long-form videos with 180s duration on average. Besides, as shown in **above Q1**, SF-LLaVA outperforms IG-VLM on LongVideoBench by over **3.5%** on average. *Third,* Our experiments are completely under the zero-shot evaluation setting. Additionally, considering that adding the Fast pathway only slightly increases the number of input tokens, we softly argue that even achieving +1% improvement on all benchmarks is not trivial. We will report all these results and discussions in the revision.

---

> > ### Author Response · Authors · 2024-11-20
> > **Response to Reviewer Hn2a (2/3)**
> >
> > **Q3:**
> > It is less well justified to keep “training free” as a requirement. It is true that training free is a nice property but existing training-based methods such as pllava and VILA involve only a lightweight fine-tuning stage that initializes mostly from image (LLaVA) model and learns only a projector or arguably simply concept selection/merging layers.
> >
> > **A3:**
> > Thank you for this question. *First,* Our paper is not intended to convince people that "training fee" is an optimal choice to achieve the best Video LLM. What is why we name our method as "a strong baseline" in the title. *Second,* lightweight model fine-tuning does not reduce the impact of training-free methods, since it still requires high computation cost. For example, fine-tuning PLLaVA-7B with LoRA takes about 35 hours on 8 A100-80G GPUs. This resource may not be afforded by many small company or university labs. We believe "training free" is beneficial for them to conduct research in Video LLM. *Third,* training multimodal LLMs is not just a computation problem. In practice, it also involves video data collection from a legally approved source, human/machine annotation, related training infra support and quality verification. Being able to use an image model out-of-box for video understanding can be very beneficial in many real-world scenarios.
> >
> > ---
> >
> > **Q4:**
> > Limited comparison with strong training based methods, such as VILA, and more comprehensive benchmarks such as MVBench or VideoMME. The benchmarks that are currently studied are either spatial biased (i.e. a 1-frame model can do quite well) or temporal-biased (EgoSchema that is based on high level semantics), rather than those that require both spatial and temporal reasoning.
> >
> > **A4:**
> > *First*, as suggested, we evaluate SF-LLaVA on VideoMME and MVBench as shown below. To compare with training-free baselines, we evaluated IG-VLM by using their official codebase. We find that SlowFast-LLaVA outperforms this strong baseline on these benchmarks by a clear margin.
> >
> > | Model | VideoMME (Average) | MVBench (Average)
> > | --- | :---: | :---:
> > | IG-VLM-7B | 39.8% | 41.5%
> > | SF-LLaVA-7B | 41.2% | 44.1%
> > | IG-VLM-34B | 50.9% | 48.1%
> > | SF-LLaVA-34B | **52.9%** | **49.8%**
> >
> > *Second,* as suggested, we compare with VILA1.5, since we didn't find any video results in the VILA paper. We use the numbers in VILA GitHub repo. The table below shows that SF-LLaVA-34B outperforms VILA1.5-40B on ActivityNet-QA, MSRVTT-QA and TGIF-QA benchmarks, and is on-par with it on MSVD-QA. In the revision, we will compare with VILA1.5 as our baseline.
> >
> > | Model | ActivityNet-QA | MSVD-QA | MSRVTT-QA | TGIF-QA
> > | --- | --- | --- | --- | ----
> > | Llama-3-VILA1.5-8B | 54.3% | 78.3% | 60.1% | 54.1%
> > | SF-LLaVA-7B | 55.5% | 79.1%  | 65.8%  | 78.7%
> > | VILA1.5-40B | 58.0% | **80.1%** | 63.0% | 58.2%
> > | SF-LLaVA-34B | **59.2%** | 79.9% | **67.4%** | **80.6%**
> >
> > ---
> >
> > **Q5:**
> > It may be interesting to see the difference in the order of input tokens. The current model looks like a bag of words model with the LLM doing most of the heavy lifting. Does the order of the tokens really matter? Does the model really learn motion? Do the datasets require/measure motion? This may give more insights to the community, if the goal is not to push the best results on video benchmarks.
> >
> > **A5:**
> > This is an excellent question. We shuffle the order of tokens before adding positional encoding and evaluate it on ActivityNet-QA and EgoSchema using SF-LLaVA-7B. The table below shows that shuffling the token order decreases the performance on both benchmarks, suggesting the motion information is important to achieve the best results. However, the performance gap on ActivityNet-QA is much smaller than EgoSchema (1.4% *v.s.* 3.6%). This is because many questions in ActivityNet-QA, such as "indoor or outdoor" and "object counting", mostly rely on spatial recognition, and using tokens as a bag of words could be sufficient to answer them. EgoSchema is proposed for complex long-form temporal reasoning, thus shuffling the order of tokens has more negative impacts on its results. Inspired this insightful suggestion, we think this experiment can be helpful to evaluate the quality of VideoQA benchmarks. We plan to test more methods, both training-free and SFT, and more benchmarks in the future work.
> >
> > | Token Order | ActivityNet-QA | EgoSchema
> > | --- | --- | ---
> > | Normal | 55.5% | 47.2%
> > | Shuffled | 54.1% | 43.6%

---

> > > ### Author Response · Authors · 2024-11-20
> > > **Response to Reviewer Hn2a (3/3)**
> > >
> > > **Q6:**
> > > Since SF-LLAVA does not require training, is it possible to compute features offline to save some memory and thus use a lot more tokens than those methods that compute vision features during inference? In this case, is it possible to show that SF-LLAVA can work as a general add-on to any model that improves inference for any model? e.g. trained on 4 frames but can infer on many more frames with SF-Model?
> > >
> > > **A6:**
> > > First, we agree that pre-computing video features can save GPU memory. We kindly remind that inputting more frames/tokens into an LLM increases its computation complexity quadratically, significantly slowing down the inference. Besides, the number of tokens is also limited by the context length of an LLM, such as 4k in Vicuna-v1.5, but this problem can be mitigated by using stronger LLMs, such as LLaMA3.
> > >
> > > Second, benefiting from the flexibility of Transformer attention mechanism, we can always input more tokens during inference to save the training computation cost. SF-LLaVA is not intentionally designed to improve the performance of "train with less and inference with more", but to effectively adapt pre-trained Image LLMs to video senarios. But we agree that this is an interesting exploration and we will evaluate its results in our future work.
> > >
> > > ---
> > >
> > > **Q7:**
> > > Does SF-LLaVA achieve good results on zero-shot generalization cases, compared with video trained methods such as pllava and vila? e.g. these methods are trained on specific datasets and therefore work well on those datasets that are similar to the training distribution, while SF-LLAVA works out of box and better on some datasets that are more different than the training sets of these existing papers.
> > >
> > > **A7:**
> > > Thank you for this question. We observe that SF-LLaVA achieves better results under the zero-shot evaluation than PLLaVA and VILA1.5.
> > >
> > > - PLLaVA-7B uses VideoChat2, which includes raw videos from ActivityNet and TGIF. Thus, its performance gap compared to SF-LLaVA-7B on these two benchmarks are relatively small (< 1%). However, on other open-ended benchmarks, such as MSRVTT, SF-LLaVA-7B outperforms PLLaVA-7B largely (+3.8%).
> > > - VILA1.5 is trained on ShareGPTVideo (including raw videos from ActivityNet), thus, its performance gap compared to SF-LLaVA is relatively small on ActivityNet-QA (about 1%). However, on TGIF-QA, where VILA1.5 was not trained on, SF-LLaVA significantly outperforms it by over +20%.
> > >
> > > We will give more discussions in the revision.
> > >
> > > ---
> > >
> > > **Q8:**
> > > Following the discussion on weakness #3, is it possible to apply SF-LLaVA on larger scale models where video training is less feasible such as a 70B/90B llama3 model? This may be a real advantage of training-free vs. training-based methods.
> > >
> > > **A8:**
> > > Thank you for this suggestion. Since we observe significant performance improvements from our 7B to 34B model, we are confident that SF-LLaVA using larger LLMs can get better results. However, SF-LLaVA-34B using batch size 1 already takes 80GB GPU memory during inference, it is not trivial for us to try larger models. Possible solutions include pre-computing the video features to save GPU memory and using distributed inference techniques. However, due to the limited time, we will leave this experiment in the future work.

---

> > > > ### Author Response · Authors · 2024-11-21
> > > >
> > > > Dear Reviewer Hn2a,
> > > >
> > > > Thank you for your insightful comments! Please let us know if our response resolved your concern. We are looking forward to your feedback!

---

> > > > > ### Author Response · Authors · 2024-11-29
> > > > >
> > > > > Dear Reviewer Hn2a,
> > > > >
> > > > > Thanks again for your valuable feedback! As we are close to the rebuttal deadline, we would like to double check with you if our response resolved your major concern and if you are willing to raise your score? Thank you!
> > > > >
> > > > > Authors of Submission610

---

### Meta-Review · Area_Chair_dwVV · 2024-12-21

**Metareview:**

The paper is borderline and received mixed reviews post-rebuttal. The overall contribution is (quoting Hn2a) it "proposes to concatenate features in a slow fast manner", and "achieves improved results on video MLLM benchmarks". The major weakness concern is on novelty, and (to AC) on the uncertainty for an improved draft. The AC checked all the materials. While appreciating the detailed responses during the rebuttal period, the AC still believes the concerns outweigh the reasons to accept the paper. The authors are encouraged to further polish the paper incorporating all the promised changes for the next cycle.

**Additional Comments On Reviewer Discussion:**

- One major concern is on novelty. Two-stream or Slow-fast design is a classical idea for video understanding, and it has been proven effective multiple times. Therefore, it is not of great surprise to see this can also work for MLLMs. Maybe one interesting enhancement is on carefully studying the improvement of this Slow-fast idea *with and without* training, which may provide further insights.
- Another potential concern is on the uncertainty of an improved draft. The AC appreciates the detailed responses from authors during the rebuttal period, but at the same time it leaves uncertainty about how the new results and changes will affect the final draft of the paper. ICLR allows major modifications to the draft during rebuttal, which will significantly reduce the uncertainty on new contents that haven't gone through reviews. While there are promises (e.g., A1 to zed3), without the actual updated draft it is significantly harder to justify for acceptance.

---

### Decision · Program_Chairs · 2025-01-22

Reject